# The Use of Computational Fluid Dynamics for Assessing Flow-Induced Acoustics to Diagnose Lung Conditions

**Khanyisani Mhlangano Makhanya** [1,*], **Simon Connell** [1], **Muaaz Bhamjee** [1,†] **and Neil Martinson** [2]

1 Department of Mechanical Engineering Science, Faculty of Engineering and the Built Environment, University of Johannesburg, Auckland Park, Johannesburg 2006, South Africa; shconnell@uj.ac.za (S.C.); muaaz.bhamjee@ibm.com (M.B.)

2 Perinatal HIV Research Unit (PHRU), University of the Witwatersrand, Johannesburg 2000, South Africa; martinson@phru.co.za

* Correspondence: kmmakhanya@gmail.com

† Current address: IBM Research—Africa, 47 Juta Street, Braamfontein, Johannesburg 2000, South Africa.

**Abstract:** Pulmonary diseases are a leading cause of illness and disability globally. While having access to hospitals or specialist clinics for investigations is currently the usual way to characterize the patient's condition, access to medical services is restricted in less resourced settings. We posit that pulmonary disease may impact on vocalization which could aid in characterizing a pulmonary condition. We therefore propose a new method to diagnose pulmonary disease analyzing the vocal and cough changes of a patient. Computational fluid dynamics holds immense potential for assessing the flow-induced acoustics in the lungs. The aim of this study is to investigate the potential of flow-induced vocal-, cough-, and lung-generated acoustics to diagnose lung conditions using computational fluid dynamics methods. In this study, pneumonia is the model disease which is studied. The hypothesis is that using a computational fluid dynamics model for assessing the flow-induced acoustics will accurately represent the flow-induced acoustics for healthy and infected lungs and that possible modeled difference in fluid and acoustic behavior between these pathologies will be tested and described. Computational fluid dynamics and a lung geometry will be used to simulate the flow distribution and obtain the acoustics for the different scenarios. The results suggest that it is possible to determine the difference in vocalization between healthy lungs and those with pneumonia, using computational fluid dynamics, as the flow patterns and acoustics differ. Our results suggest there is potential for computational fluid dynamics to enhance understanding of flow-induced acoustics that could be characteristic of different lung pathologies. Such simulations could be repeated using machine learning with the final objective to use telemedicine to triage or diagnose patients with respiratory illness remotely.

**Keywords:** computational fluid dynamics; CFD; flow-induced acoustics; pulmonary disease; vocalization

## 1. Introduction

Lungs are vulnerable internal organs as they are exposed to the environment through inhaled ambient air, which potentially carries pollution and infectious particles. Respiratory diseases cause a substantial burden on individuals, communities, and governments, with tuberculosis (TB), pneumonia, and asthma being common in South Africa [1]. The health burden imposed by respiratory diseases has led to a realization that there is a growing need for reliable and rapid diagnostic devices, that could differentiate between severe and mild respiratory illness and even rapidly diagnose pulmonary conditions. Herein, we use CFD methods to model the flow-induced vocal-, cough-, and lung-generated acoustics to assess the potential for acoustic diagnosis of lung pathology. This paper investigates the acoustics propagated in the lungs by comparing the surface acoustic power of healthy and diseased lungs. The sound power level is a measure of the rate at which acoustic energy is radiated

from a source and the sound intensity arial density measures the acoustic power passing through a unit area [1].

Computational fluid dynamics (CFD) is the use of numerical analysis and data structure methods to solve fluid-related problems [2]. The use of CFD is emerging in biomedicine with the potential development of new and improved diagnostic and therapeutic devices. Although there has been progress made with the use of CFD in the biomedical field, the difficulty has been in understanding the complexity of the human anatomy and human body fluid behavior [3]. Computational simulations provide valuable information that would otherwise be difficult to obtain experimentally. The use of CFD methods have been employed to analyse the airflow in both healthy and infected conducting airways and lungs. Recent studies have focused on the transfer of pollutants and drug delivery within the respiratory tree using CFD [4,5]. Sleep-disordered breathing including sleep apnoea has also been a focus area in the recent literature relating to this subject [6]. During breathing, airflow from the mouth or nose into the pharynx, trachea, and bronchae can be either laminar, transitional, or turbulent. The flow boundary conditions, and the geometry of the individual are factors that affect airflow throughout its path to the most distant parts of the airway, the alveoli, where gaseous exchange occurs [7]. Advances in medical imaging allow the various geometric and anatomical features of the respiratory tract to be accurately reconstructed from scanned images and digitally converted into data which are then included in CFD models. Infections that occur in the lungs are also detected by examining vocalization and aeroacoustics, which is the branch within CFD that applies to this study. Vocalization is any sound produced through the respiratory system and it can indicate information about the respiratory condition of the individual, as auscultation is the most effective and simple method to detect abnormality of the lung respiration invasively [8,9]. The noise generated within the lungs would be both impulsive and turbulent. Impulsive noise is a result of moving surfaces or surfaces in nonuniform flow conditions. Within the lungs there is surface interaction between the fluid and the lung walls which therefore creates an impulsive noise source.

A key parameter used in the study is the peak expiratory flow rate which differs for healthy and infected lungs. The volumetric flow rate in the lungs can be determined from the peak expiratory flow rate. The volumetric flow rate in healthy lungs is higher than the volumetric flow rate in infected lungs. The studies by Patil et al. and Sitalakshmi et al. propose that the peak expiratory flow rate inside healthy lungs is between 7.5 L/s and 5 L/s (from 450 L/min to 300 L/min), and inside infected lungs it is between 2.7 L/s and 1.6 L/s (from 162 L/min to 96 L/min) [10,11].

We will model flow-induced vocal-, cough-, and lung-generated acoustics to diagnose lung conditions using computational fluid dynamics methods. The objectives include producing CFD models with realistic conditions for a healthy and infected lung (consolidated), modeling and assessing the airflows behavior in healthy and infected lungs, and predicting vocal changes when varying the extent of pulmonary consolidation.

## 2. Materials and Methods

### 2.1. Model Geometry and Mesh

We made use of lung geometry received from North Carolina State University [12], which is a patient-specific model based on a previous study by Su and Cheng [13]. The lung geometry consists of four sections—namely, the oropharynx, larynx, trachea, and bronchus. A tube extension was included on the geometry to allow for CFD stability as the fluid enters the oral cavity. The model uses bronchial bifurcations to the 3rd generation, with 10 respiratory bronchioles. The mesh type used is a tetrahedron/mixed mesh with a cell size of 3 mm and a total cell count of $10.1 \times 10^6$ cells. A mesh independent study was performed to verify whether the cell size used is suitable and convergence is achieved. Table 1 below indicates the different cell sizes assessed to determine mesh independence and the asymptotic range calculated based on the surface acoustic power. The grid convergence index (GCI) method at mesh sizes (3 mm, 6 mm, 12 mm) was used for three different positions

which are based on the performance parameter being the surface acoustic power on the larynx [14]. As the asymptotic range is close to 1 for all 3 positions, mesh independence is achieved on the 3 mm (~10 million cell mesh). Thus, the results presented are based on the 3 mm mesh.

**Table 1.** GCI Analysis for Surface Acoustic Power—Mesh Independence Study.

| Position | $f_1$ | $f_2$ | $f_3$ | $GCI_{12}$ | $GCI_{23}$ | Asymptotic Range |
|---|---|---|---|---|---|---|
| Position 1 | $6.5 \times 10^{-28}$ | $6.4 \times 10^{-28}$ | $5.8 \times 10^{-28}$ | 0.003841615 | 0.0234375 | 1.016 |
| Position 2 | $6.6 \times 10^{-28}$ | $6.38 \times 10^{-28}$ | $5.6 \times 10^{-28}$ | 0.01636905 | 0.06003695 | 1.034 |
| Position 3 | $6.4 \times 10^{-28}$ | $6.3 \times 10^{-28}$ | $5.7 \times 10^{-28}$ | 0.00390625 | 0.02380952 | 1.016 |

A time step study was performed. The mesh cell size of 3 mm was kept constant during this study and the time step were varied (1 s, 0.1 s, 0.01 s, 0.001 s). During the study, convergence was achieved for all the time steps except at 1 s. The grid convergence index (GCI) method at the time steps (1 s, 0.1 s, 0.01 s, 0.001 s) was used based on the performance parameter being the surface acoustic power on the larynx. Table 2 indicates the grid convergence index (GCI) at two different positions on the larynx. The asymptotic range is close to 1 for both the positions.

**Table 2.** GCI Analysis for the Surface Acoustic Power—Time Step Study.

| Position | $f_1$ | $f_2$ | $f_3$ | $GCI_{12}$ | $GCI_{23}$ | Asymptotic Range |
|---|---|---|---|---|---|---|
| Position 1 | $2.36 \times 10^{-27}$ | $2.40 \times 10^{-27}$ | $2.40 \times 10^{-27}$ | 0.025207785 | 0.001032223 | 0.981 |
| Position 2 | $7.39 \times 10^{-28}$ | $7.81 \times 10^{-28}$ | $7.86 \times 10^{-28}$ | 0.081109195 | 0.00851182 | 0.945 |

Table 3 below indicates the results obtained for the different time steps (1 s, 0.1 s, 0.01 s, 0.001 s) which use the CFL condition. Convergence was obtained for the time steps (0.1 s, 0.01 s, 0.001 s). In Table 3, the surface acoustic power at an identical position on the larynx for the different time steps (1 s, 0.1 s, 0.01 s, 0.001 s) is also indicated. From the time step 0.1 s and decreasing by a magnitude of 10, there is no significant change in the performance parameter (surface acoustic power). From the results obtained, 3 mm mesh size with a time step size of 0.1 s is both mesh and time step size independent.

**Table 3.** Time Step Study.

| Time Step (s) | $\Delta x$ (m) | Velocity Magnitude | CFL | Surface Acoustic Power (W/m$^2$) | Convergence Achieved |
|---|---|---|---|---|---|
| 1 | | | 0.5333 | $2.00 \times 10^{-27}$ | No |
| 0.1 | 0.003 | 0.0016 | 0.0533 | $2.36 \times 10^{-27}$ | Yes |
| 0.01 | | | 0.0053 | $2.40 \times 10^{-27}$ | Yes |
| 0.001 | | | 0.0005 | $2.40 \times 10^{-27}$ | Yes |

The Figure 1 below indicates the lung geometry used and a section of the lung geometries mesh.

### 2.2. CFD Procedures and Boundary Conditions

The lung geometry model received from North Carolina State University was an unstructured mesh (.uns file) and was imported into ICEM CFD in ANSYS WorkBench 2021 (ANSYS Inc., Canonsburg, PA, USA) and named as the complete model. ANSYS ICEM CFD is a tool used to manipulate the geometry and perform meshing applications. It was designed to mainly import complicated geometries. Tetrahedron elements are employed to mesh the model and there are a total of $10.1 \times 10^6$ elements and $2.5 \times 10^6$ total nodes. ICEM CFD allows for mesh generation with the ability to compute meshes

with different structures. Fluid Flow (Fluent) was thereafter selected on Workbench and a connection is made between ICEM CFD and Fluid Flow (Fluent). In Fluid Flow (Fluent), the acoustics model selected using a broadband noise source and the viscous model selected is the k-epsilon. The fluid material description is air at a density of 1.225 kg/m$^3$ and the solid material representative for the adventitia (outer membrane) is polyurethane with a density of 949.79 kg/m$^3$ [15]. Rubber polyurethane is used as the density as this is a close representative of the adventitia (outer membrane). For the boundary conditions specification, the 10 bronchioles are set as the inlet for air flow and the oral cavity is set as the air flow outlet. The volumetric flow rate is set for four different measurements, a healthy lung at 5 L/s and 7.5 L/s and an infected lung at 1.6 L/s and 2.7 L/s. A hybrid initialization is implemented. Table 4 below shows a summary of the material properties and boundary conditions [16].

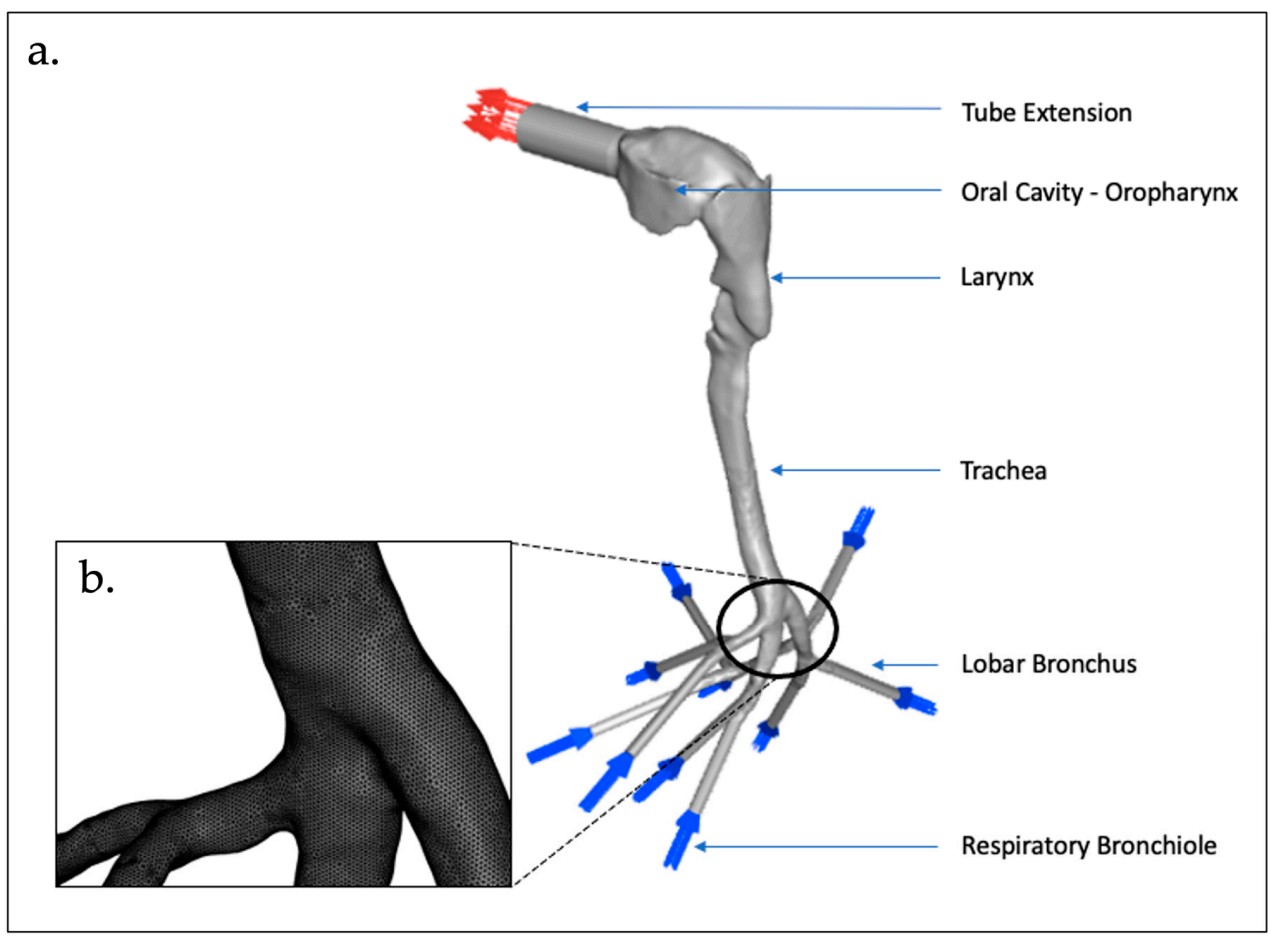

**Figure 1.** Representative Lung Geometry: (**a**) lung geometry with labelled sections; and (**b**) mesh.

**Table 4.** Material Properties and Boundary Conditions [1].

| Material Properties | | |
|---|---|---|
| Fluid Material | Air | Density—1.225 kg/m$^3$ |
| Solid Material—Adventitia (Outer Membrane) | Representative Material | Density—949.79 kg/m$^3$ |
| **Boundary Conditions** | | |
| Boundary | Type | Condition |
| Oral Cavity | Pressure Outlet | Atmospheric Pressure |
| Respiratory Bronchiole | Velocity Inlet | Case Dependent |
| Wall (Oropharynx, Larynx, Trachea, Lobar Bronchus) | Wall Boundary | No Slip |

**Table 4.** *Cont.*

| Solver Settings | |
| --- | --- |
| Number of Time Steps | 100 |
| Time Step Size (s) | 0.1 |
| Max Iterations/Time Step | 20 |
| Pressure Velocity Coupling—Type | Simple |
| Discretization Scheme—Pressure | 2nd Order |
| Discretization Scheme—Momentum | 2nd Order Upwind |
| Discretization Scheme—Turbulent Kinetic Energy | 1st Order Upwind |
| Discretization Scheme—Turbulent Dissipation Rate | 1st Order Upwind |

*2.3. Governing Equations*

2.3.1. CFD Model—Broadband Noise Source Models

Proudman, using Lighthill's acoustic analogy, derived a formula for acoustic power which is generated by isotropic turbulence without mean flow. Lilley rederived the formula by accounting for the retarded time difference. Both the derivations yield acoustic power due to the unit volume of isotropic turbulence in in W/m$^3$ as [17]:

$$P_A = \alpha \rho_0 \left( \frac{u^3}{l} \right) \frac{u^5}{a_0^5} \tag{1}$$

where $u$ and $l$ are the turbulence velocity and length scales, respectively, and $a_0$ is the speed of sound. $\alpha$ in Equation (1) is a model constant. In terms of $k$ and $\varepsilon$, Equation (1) can be written as [17]:

$$P_A = \alpha_\varepsilon \rho_0 \varepsilon M_t^5 \tag{2}$$

where,

$$M_t = \frac{\sqrt{2k}}{a_0} \tag{3}$$

The rescaled constant, $\alpha_\varepsilon$, is set to 0.1 in ANSYS FLUENT based on the calibration of Sarkar and Hussaini using direct numerical simulation of isotropic turbulence. ANSYS FLUENT can also report the acoustic power in dB, which is computed from:

$$L_p = 10 \log \left( \frac{P_A}{P_{ref}} \right) \tag{4}$$

$P_{ref}$ is the reference acoustic power $P_{ref} = 10^{-12}$ W/m$^3$

Proudman's formula gives an approximate measure of the local contribution to total acoustic power per unit volume in a given turbulence field. Proper caution, however, should be taken when interpreting the results in view of the assumptions made in the derivation, such as high Reynolds number, small Mach number, isotropy of turbulence, and zero mean motion [17].

2.3.2. CFD Model—Transport Equations

The realizable $k$-$\varepsilon$ model differs from the standard $k$-$\varepsilon$ model. The $k$-$\varepsilon$ model satisfies certain mathematical constraints on the Reynolds stresses consistent with the physics of turbulent flows. The modeled transport equations for $k$ and $\varepsilon$ in the realizable $k$-$\varepsilon$ model are indicated below [17]:

$$\frac{\partial}{\partial t}(\rho k) + \frac{\partial}{\partial x_j}(\rho k u_j) = \frac{\partial}{\partial x_j}\left[ \left( \mu + \frac{\mu_t}{\sigma_k} \right) \frac{\partial k}{\partial x_j} \right] + G_k + G_b - \rho \epsilon - Y_M + S_k \tag{5}$$

$$\frac{\partial}{\partial t}(\rho\varepsilon) + \frac{\partial}{\partial x_j}(\rho\varepsilon u_j) = \frac{\partial}{\partial x_j}\left[\left(\mu + \frac{\mu_t}{\sigma_\epsilon}\right)\frac{\partial\epsilon}{\partial x_j}\right] + \rho C_1 S\varepsilon - \rho C_2 \frac{\varepsilon^2}{k + \sqrt{v\epsilon}} + C_{1\epsilon}\frac{\epsilon}{k}C_{3\epsilon}G_b + S_\epsilon \quad (6)$$

where,

$$C_1 = max\left[0.43, \frac{\eta}{\eta + 5}\right], \ \eta = S\frac{k}{\epsilon}, \ S = \sqrt{2S_{ij}S_{ij}} \quad (7)$$

In the above equations, $G_k$ represents the generation of turbulence kinetic energy as a result of the mean velocity gradients. $G_b$ represents the generation of the turbulence kinetic energy as a result of the buoyancy. $Y_M$ represents the contribution of the fluctuating dilation in compressible turbulence to the overall dissipation. $C_2$ and $C_{1\varepsilon}$ are constraints. $\sigma_k$ and $\sigma_\varepsilon$ are the turbulent Prandtl numbers for $k$ and $\varepsilon$, respectively. $S_k$ and $S_\varepsilon$ are user-defined source terms.

The Continuity Equation is,

$$\frac{\partial\rho}{\partial t} + \nabla\cdot(\rho\overline{u}) = 0 \quad (8)$$

$$\rho\frac{\partial\overline{u}_i}{\partial t} + \rho\nabla\cdot(\overline{u}_i\overline{u}) = -\frac{\partial\overline{\rho}}{\partial\overline{x}_i} + \nabla\cdot(\mu\nabla\overline{u}_i) - B_i - \rho\frac{\partial}{\partial x_i}\left(\overline{u'_i u'_j}\right) + S_M \ \ i,j,l = x,y,z \quad (9)$$

$$u_i = \overline{u}_i + u'_i \quad (10)$$

The Momentum Equation is,

$$\frac{\partial}{\partial t}\left(\rho\vec{v}\right) + \nabla\cdot\left(\rho\vec{v}\vec{v}\right) = -\nabla p + \nabla\cdot\left[\mu\left(\nabla\vec{v} + \nabla\vec{v}^T\right)\right] + \rho\vec{g} + \vec{F} \quad (11)$$

The Boussinesq hypothesis to relate to the Reynolds stresses to the mean velocity gradients [17],

$$-\rho\left(\overline{u'_i u'_j}\right) = \mu_t\left(\frac{\partial u_i}{\partial x_j} + \frac{\partial u_j}{\partial x_i}\right) - \frac{2}{3}\left(\rho k + u_t\frac{\partial u_k}{\partial x_k}\right)\delta_{ij} \quad (12)$$

### 2.4. Data Analysis and Comparison

Based on the CFD simulations, the results were plotted as scatter plots indicating the total pressure in pascal (Pa) versus the surface acoustic power in watts per cubic meter (W/m$^2$). The obtained surface acoustic power was thereafter used to calculate the arial density of the sound intensity which can be expressed relative to the reference intensity which is the threshold of hearing at $10^{-12}$ W/m$^2$. The equation is expressed in a logarithmic decibel scale as

$$L_I = 10 \ log_{10}(I/I_{ref}) \quad (13)$$

$$L_I = 10 \ log_{10}(I) + 120 \quad (14)$$

$$I = 10^{\frac{L_I - 120}{10}} \quad (15)$$

where
　　$L_I$ = Sound Intensity Level (decibel, dB)
　　$I$ = Sound Intensity (W/m$^2$)
　　$I_{ref}$ = $10^{-12}$—Reference Sound Intensity (W/m$^2$).

## 3. Results and Discussion

### 3.1. Validation of Simulated Flow Velocity

Figures 2–6 show the results for the surface acoustic power as a function of total pressure (Pa) for healthy and infected lungs. Figures 7–11 show the results for the total pressure in pascal (Pa) versus the sound intensity level in decibels (dB). Results for the sound intensity level of a normal healthy lung were based on results from Oliveira et al., Gavriely et al., and Pasterkamp et al. [18–20]. It is noted that the sound intensity level for a normal healthy lung during expiration is between 5 and 20 dB.

### 3.2. Surface Acoustic Power for Healthy Lungs and Infected Lungs at the Larynx

The first graphs that were plotted are the surface acoustic power (W/m²) which is an arial power density as a function of total pressure (Pa) for healthy (7.5 L/s and 5 L/s) and infected lungs (2.7 L/s and 1.6 L/s) at the larynx. There were four separate simulations run for the same lung geometry model at different flow rates. The first 2 simulations were for healthy lungs at an expiratory flow rate of 7.5 L/s (450 L/min) and 5 L/s (300 L/min), respectively [10,15,21]. The final 2 simulations were run for infected lungs operating at an expiratory flow rate of 2.7 L/s (162 L/min) and 1.6 L/s (96 L/min), respectively, based on prior reports [10,15,21].

Figure 2 below indicates the surface acoustic power as a function of total pressure for healthy and infected lungs. The results indicated in Figure 2 show that the surface acoustic power was the highest for the healthy lungs at an expiratory flow rate of 7.5 L/s (450 L/min) and the lowest for the infected lungs at an expiratory flow rate of 1.6 L/s (96 L/min). The results for the four simulations are bell-shaped which indicates that at a specific point where the total pressure acts on the larynx, the surface acoustic power reaches a peak and then steadily decreases. The results obtained indicate that although the same region of the model, the larynx, is the reference point for measuring the results, there is a difference in the surface acoustic power as the expiratory flow rate is changed. The implications from the results are that medical practitioners would observe different acoustic measurements for patients with healthy lungs compared to patients with infected lungs.

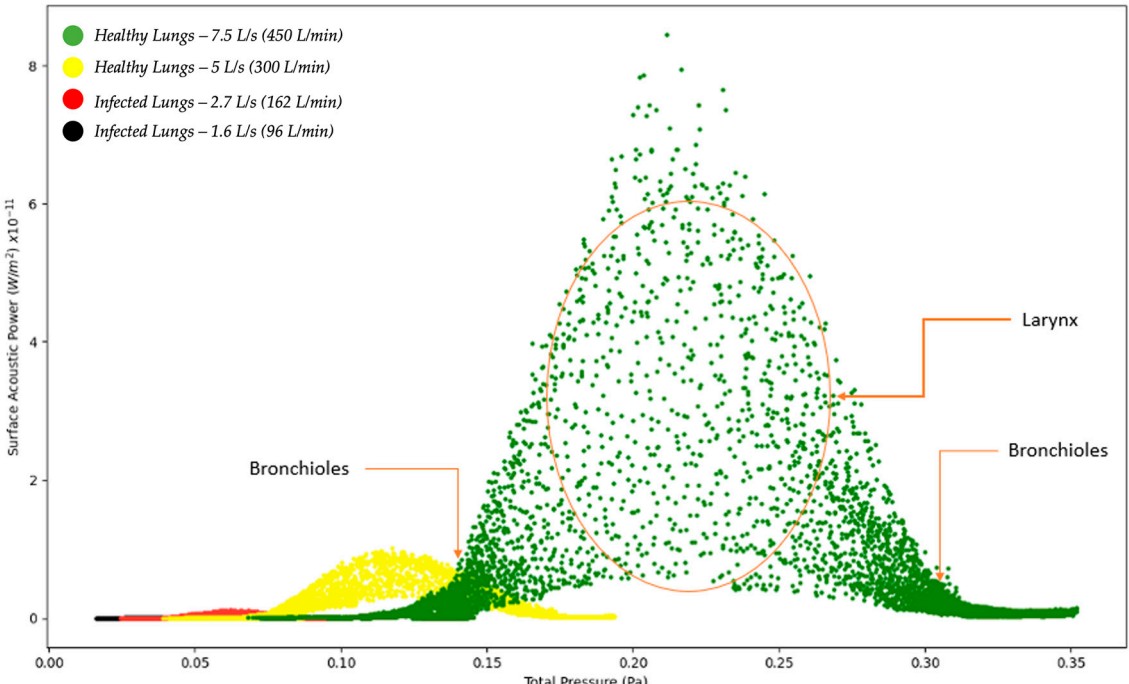

**Figure 2.** Surface Acoustic Power as a function of Total Pressure—Healthy and Infected Lungs.

Figures 3–6 below show similar results as those obtained in Figure 2; however, for different axes. The results on Figures 3–6 show the four simulations on their own scatter plot graph which also indicates the bell-shape and the peak at a specific total pressure.

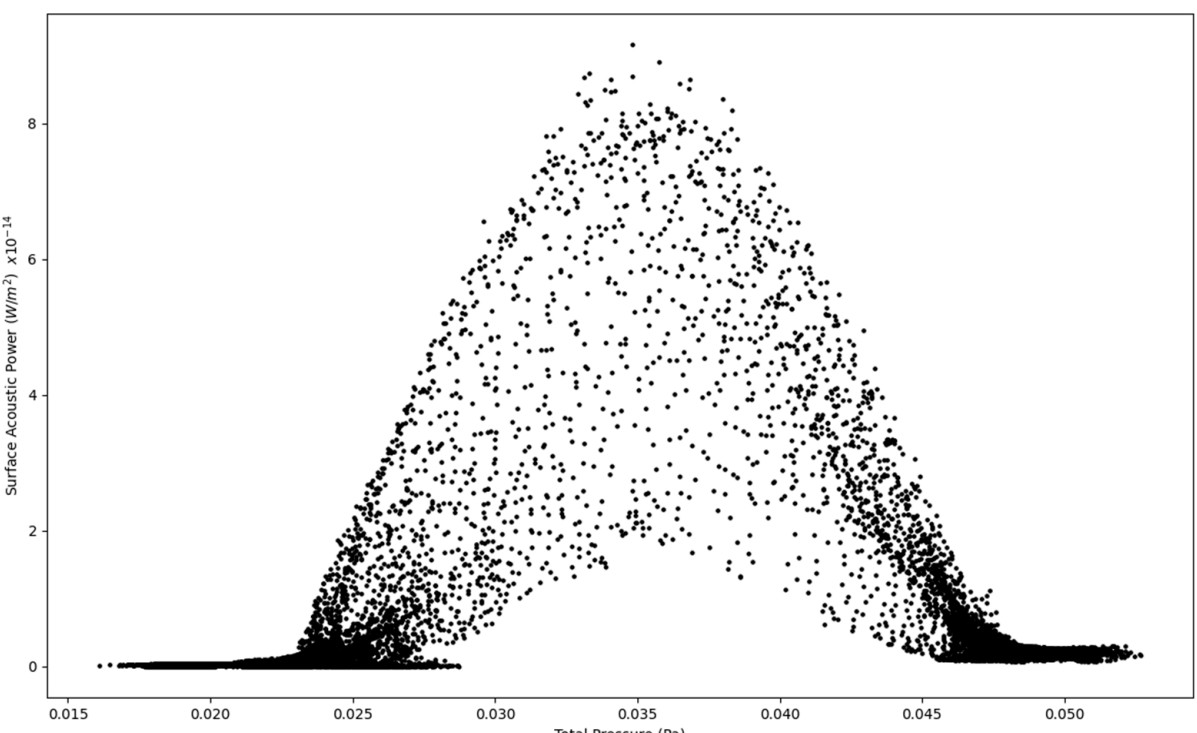

**Figure 3.** Surface Acoustic Power as a function of Total Pressure at 1.6 L/s—Infected Lung.

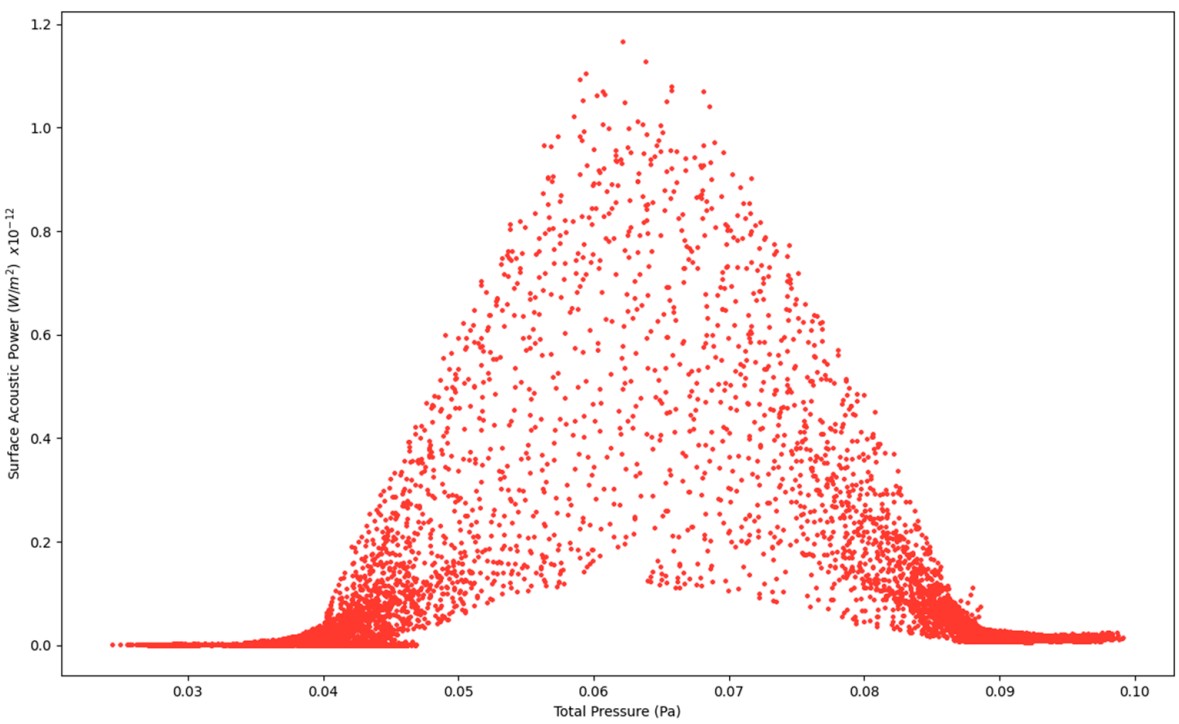

**Figure 4.** Surface Acoustic Power as a function of Total Pressure at 2.7 L/s—Infected Lung.

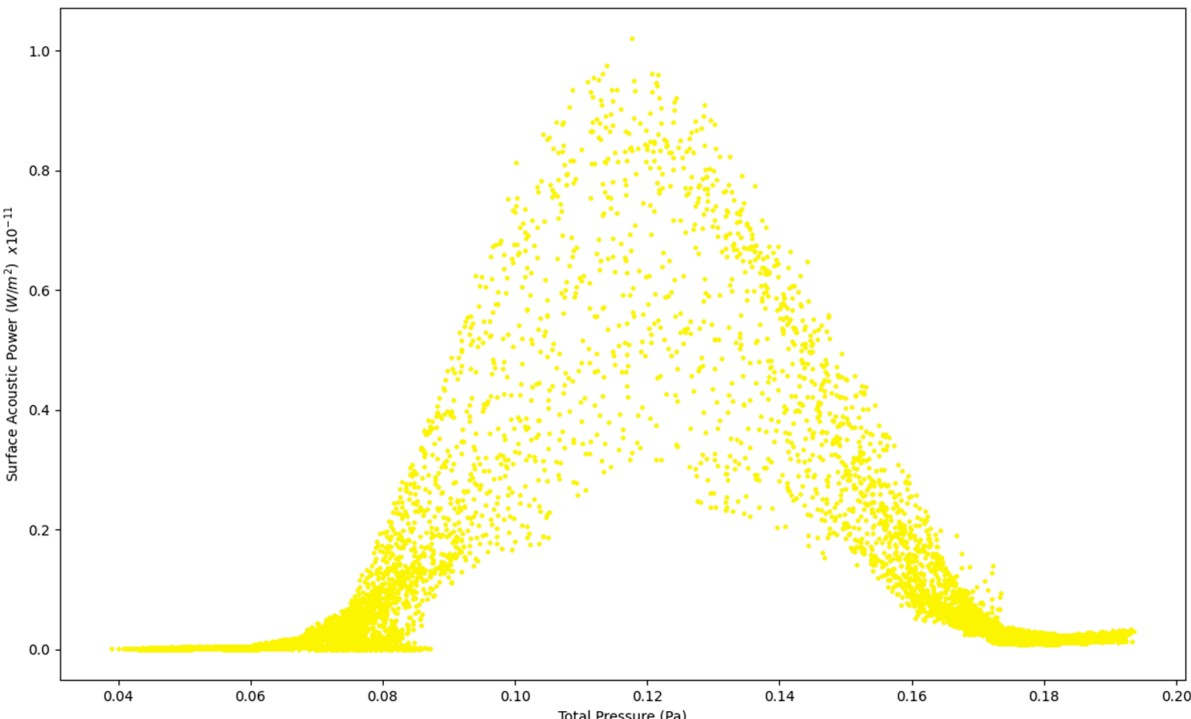

**Figure 5.** Surface Acoustic Power as a function of Total Pressure at 5 L/s—Healthy Lungs.

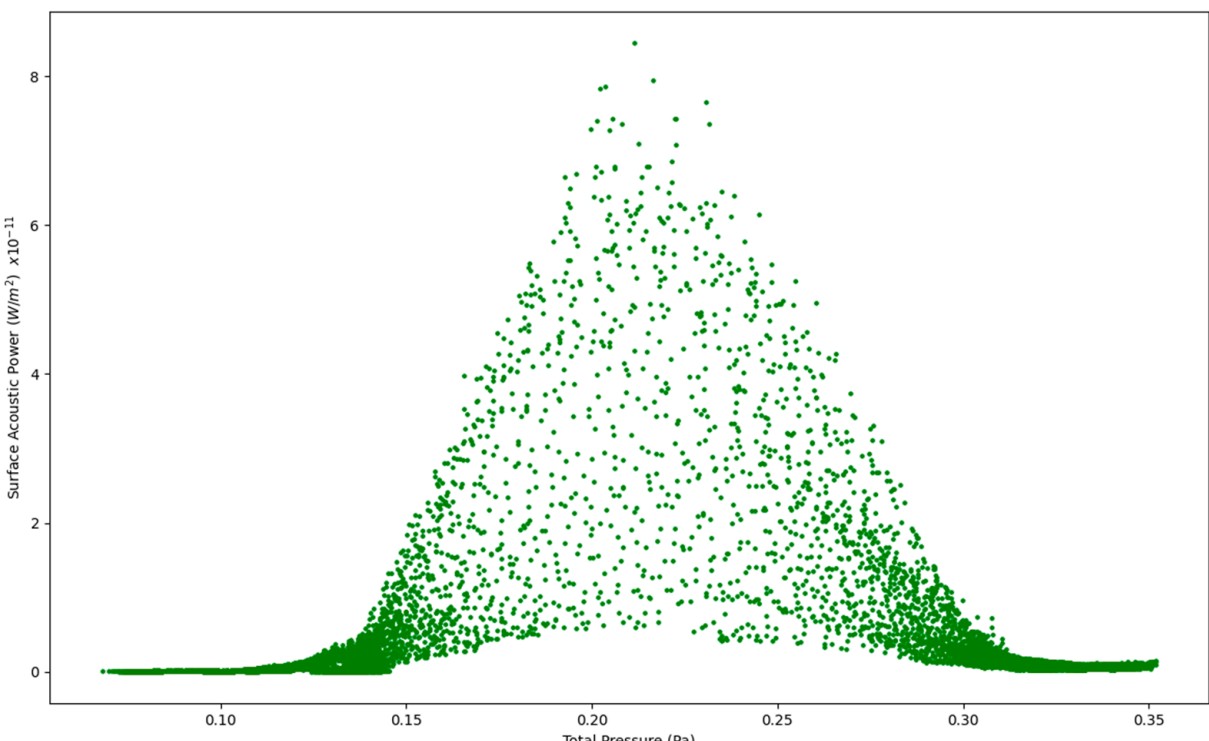

**Figure 6.** Surface Acoustic Power as a function of Total Pressure at 7.5 L/s—Healthy Lungs.

### 3.3. Sound Intensity Level for Healthy Lungs and Infected Lungs at the Larynx

The sound intensity level (dB) as a function of total pressure (Pa) for healthy and infected lungs at the larynx was modeled over four separate simulations run for the same lung geometry model at different flow rates. As in the previous section, the first 2 simulations were for healthy lungs at an expiratory flow rate of 7.5 L/s (450 L/min) and

5 L/s (300 L/min), respectively [10,15,21]. The final 2 simulations were run for infected lungs operating at an expiratory flow rate of 2.7 L/s (162 L/min) and 1.6 L/s (96 L/min), respectively, based on prior studies [10,15,21].

Figure 7 below indicates the sound intensity level as a function of total pressure for healthy and infected lungs. The results indicated in Figure 7 show that the sound intensity level was the highest for the healthy lungs at an expiratory flow rate of 7.5 L/s (450 L/min) and the lowest for the infected lungs at an expiratory flow rate of 1.6 L/s (96 L/min). The sound intensity level for the healthy lungs between 5 L/s and 7.5 L/s indicate a peak sound intensity between 5 and 20 dB [18]. The results obtained indicate that although the same region of the model, the larynx, is the reference point for measuring the results, there is a difference in the sound intensity level as the expiratory flow rate is changed. The implications from the results are that medical practitioners can obtain a range of the sound intensity level which would be able to indicate if a patient has healthy lungs or encountering a lung pathology.

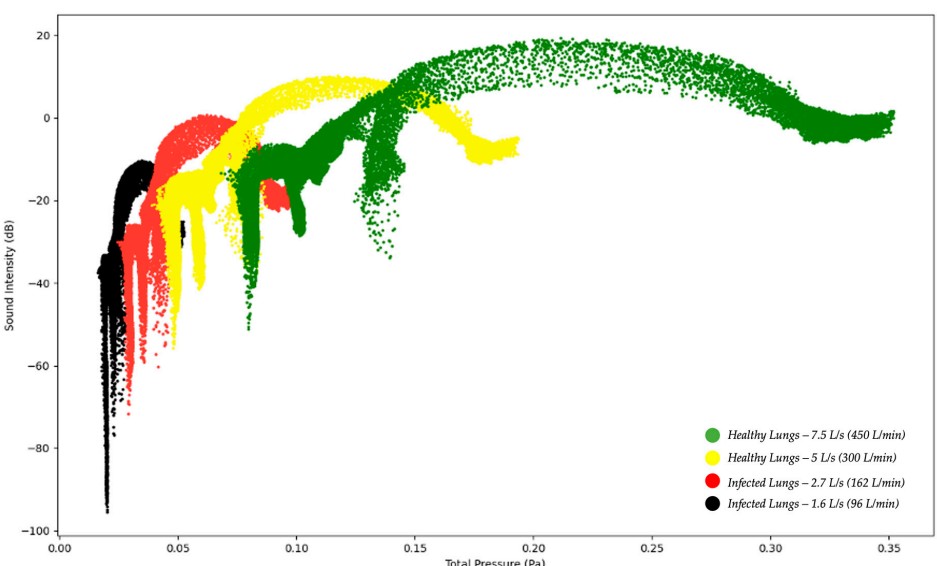

**Figure 7.** Sound Intensity Level as a function of Total Pressure—Healthy and Infected Lungs.

The results in Figures 8–11 show the four simulations on their own scatter plot graph which also indicates the highest point of the sound intensity level at a specific total pressure.

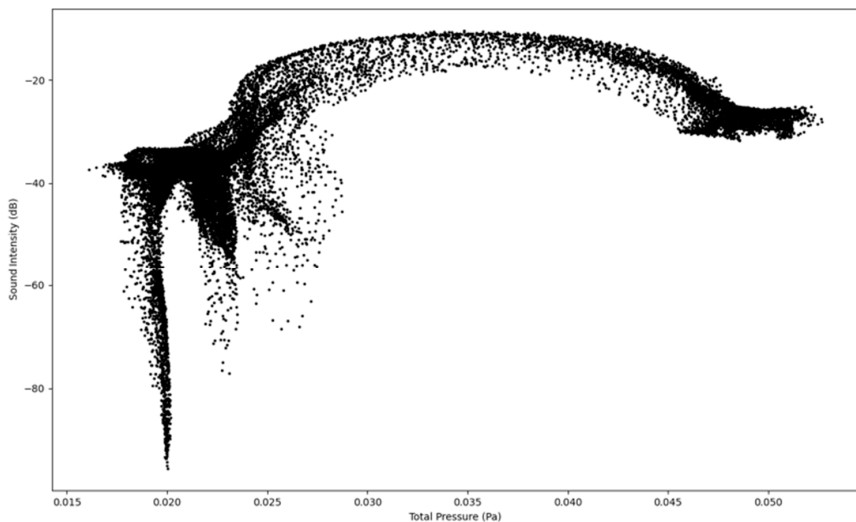

**Figure 8.** Sound Intensity as a function of Total Pressure at 1.6 L/s—Infected Lung.

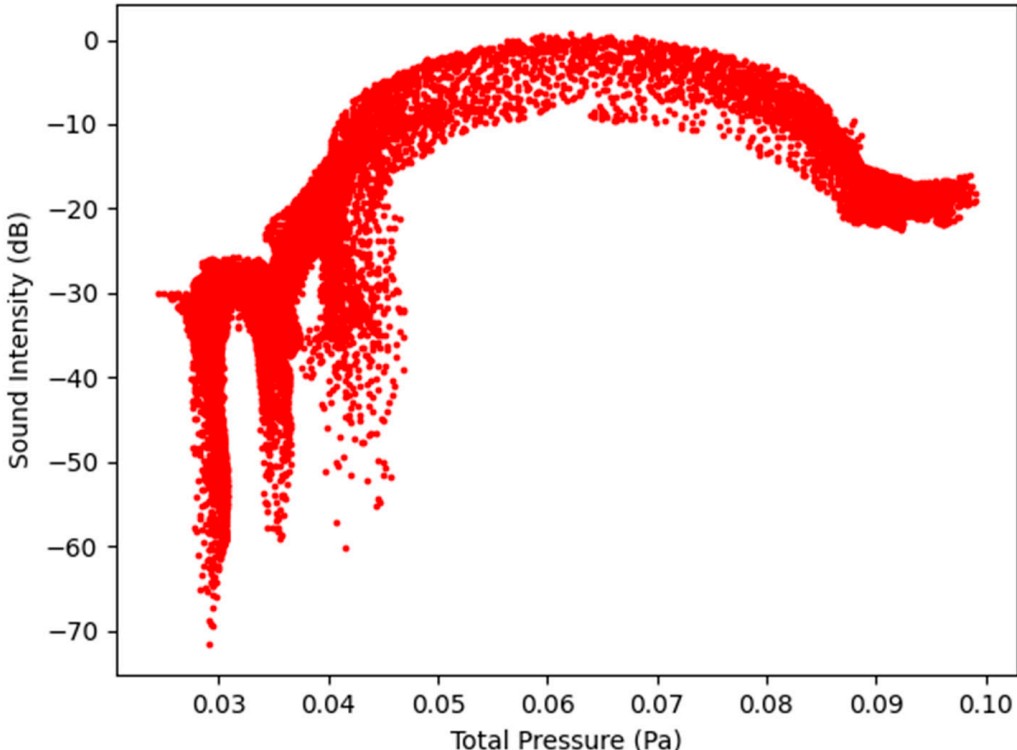

**Figure 9.** Sound Intensity as a function of Total Pressure at 2.7 L/s—Infected Lung.

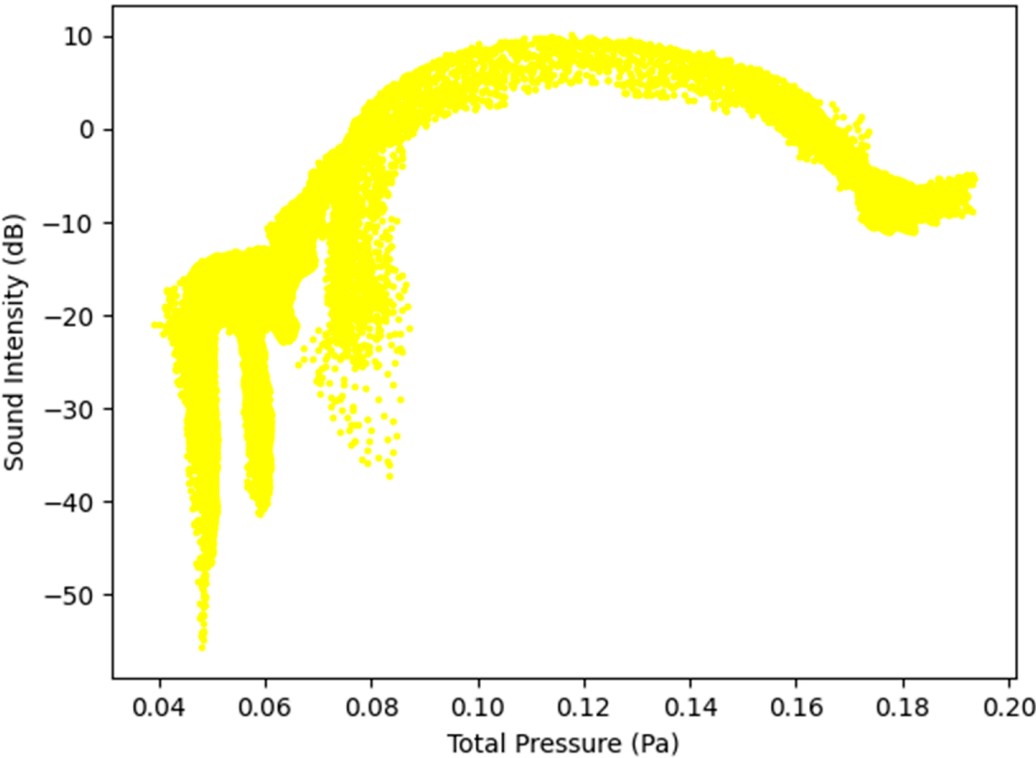

**Figure 10.** Sound Intensity as a function of Total Pressure at 5 L/s—Healthy Lungs.

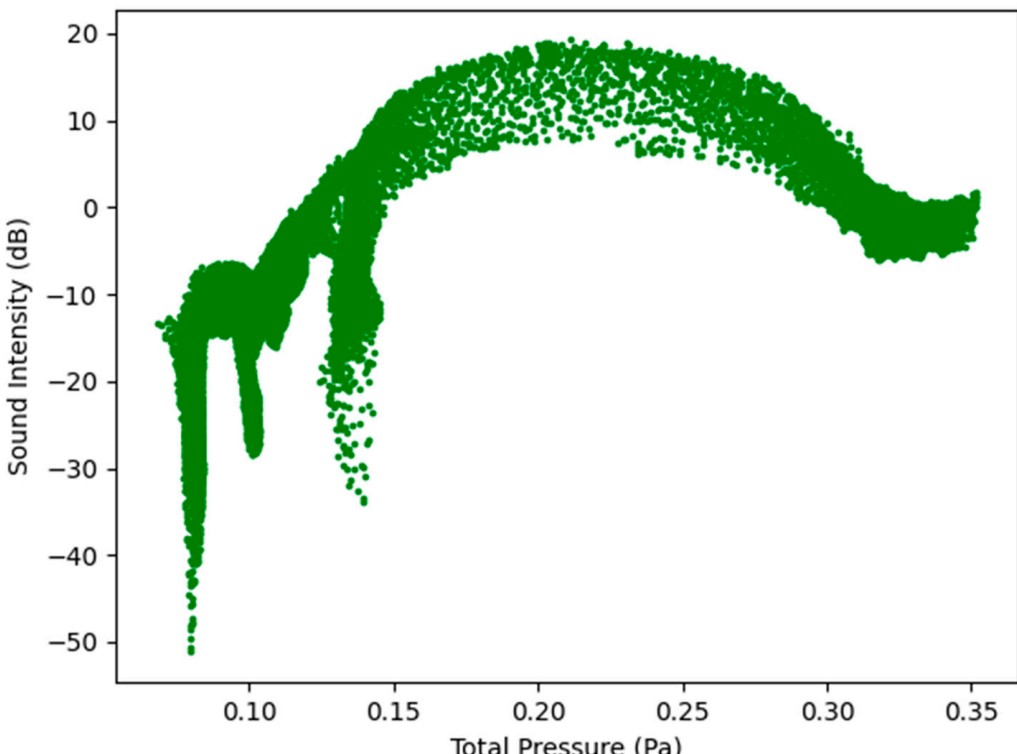

**Figure 11.** Sound Intensity as a function of Total Pressure at 7.5 L/s—Healthy Lungs.

### 4. Conclusions

This study was performed using a lung geometry model, modified with the additional tube extension to allow for CFD stability as the fluid entered the oral cavity. The bifurcation of the model was up the third generation. The study considered only expiration with the simulations performed for two healthy and two infected lung parameters. The results obtained in the study show that it is possible to determine the difference in auditory characteristics of vocalization using computational fluid dynamics between healthy and infected lungs as the flow patterns and acoustics for these pathologies differ. These results also suggest that there is the potential for the medical field to make use of computational fluid dynamics to understand the flow-induced acoustics for different lung pathologies. Therefore, the study establishes the possibility that CFD modeling can contribute to the diagnosis of pulmonary conditions. This is consistent with some clinical perceptions.

The next steps would be to couple such simulations with machine learning to improve telemedicine and diagnose pulmonary diseases remotely. There is also further potential in this study as different parameters can be used to determine the actual pulmonary disease experienced by a patient.

**Author Contributions:** Conceptualization, methodology, modeling validation, formal analysis, data curation, writing—original draft preparation, visualization, project administration by K.M.M.; Conceptualization, methodology, writing—review, editing, project administration by S.C., M.B., and N.M.; supervision by S.C., M.B., and N.M. All authors have read and agreed to the published version of the manuscript.

**Funding:** The Article Processing Charge (APC) was funded by the University of Johannesburg (UJ).

**Data Availability Statement:** Data supporting reported results can be obtained by contacting the corresponding author. The lung geometry was obtained from North Carolina State University (NC State) and permission was granted by NC State to use the geometry solely for this study and can therefore not be shared by the authors.

**Acknowledgments:** The authors acknowledge the Centre for High Performance Computing (CHPC), South Africa, for providing computational resources to this research project. Sincere gratitude is expressed to Clement Kleinstreuer and Sriram Vasudevan Chari from the Department of Mechanical Engineering, North Carolina State University for providing the lung geometry. The opinions expressed in the manuscript and the conclusions arrived at are solely those of the authors and not the University of Johannesburg (UJ), Perinatal HIV Research Unit (PHRU), North Carolina State University, the CHPC or International Business Machines (IBM) Corporation.

**Conflicts of Interest:** The authors declare no conflict of interest.

## Abbreviations/Nomenclature

The following abbreviations/nomenclature are used in this manuscript:

| | |
|---|---|
| CFD | Computational Fluid Dynamics |
| CHPC | Centre for High Performance Computing |
| dB | Decibel |
| IBM | International Business Machines |
| ICEM | Integrated Computer-aided Engineering and Manufacturing |
| INC | Incorporated |
| LES | Large Eddy Simulation |
| NCSU | North Carolina State University |
| PHRU | Perinatal HIV Research Unit |
| PU | Polyurethane |
| RANS | Reynolds Averaged Navier-Stokes |
| TB | Tuberculosis |
| USA | United States of America |
| $\vec{a}$ | Acceleration (m/s$^2$) |
| $a$ | Local speed of sound (m/s) |
| $C_D$ | Drag coefficient, defined different ways (dimensionless) |
| $c_p$, $c_v$ | Heat capacity at constant pressure, volume (J/kg-K) |
| $d$ | Diameter; $d_p$, $D_p$ particle diameter (m) |
| $\vec{F}$ | Force vector (N) |
| $F_D$ | Drag force (N) |
| $\vec{g}$ | Gravitational acceleration (m/s$^2$); standard values = 9.80665 m/s$^2$ |
| $G_k$ | Generation of turbulence kinetic energy due to the mean velocity gradients |
| $G_b$ | Generation of turbulence kinetic energy due to the buoyancy |
| $I$ | Sound Intensity (W/m$^2$) |
| $I_{ref}$ | Reference Sound Intensity (W/m$^2$) |
| $k$ | Kinetic energy per unit mass (J/kg) |
| $k$ | Reaction rate constant, e.g., k$_1$, k$_{-1}$, k$_{f,r}$, k$_{b,r}$ (units vary) |
| $k_B$ | Boltzmann constant (1.38 × 10$^{-23}$ J/molecule-K) |
| $k$, $k_c$ | Mass transfer coefficient (units vary); also $K$, $K_c$ |
| $l$, $L$ | Length scale (m, cm) |
| $L_p$ | Sound Pressure (dB) |
| $m$ | Mass (g, kg) |
| $M$ | Mach number $\equiv$ ratio of fluid velocity magnitude to local speed of sound (dimensionless) |
| $p$ | Pressure (Pa, atm, mm Hg) |
| $P_A$ | Acoustic power due to the unit volume of isotropic turbulence (W/m$^3$) |
| $P_{ref}$ | Reference Acoustic Power (W/m$^3$) |
| $r$ | Radius (m) |
| $Re$ | Reynolds number $\equiv$ ratio of inertial forces to viscous forces (dimensionless) |
| $s$ | Species entropy; s$^0$, standard state entropy (J/kgmol-K) |
| $S_{i,j}$ | Mean rate-of-strain tensor (s$^{-1}$) |

| | |
|---|---|
| T | Temperature (K) |
| t | Time (s) |
| u | Turbulence Velocity |
| V | Volume ($m^3$) |
| $Y_M$ | Contribution of the fluctuating dilation in compressible turbulence to the overall dissipation |
| $\vec{v}$ | Overall velocity vector (m/s) |
| $\propto$ | Thermal diffusivity ($m^2/s$) |
| $\propto$ | Volume fraction (dimensionless) |
| $\beta$ | Coefficient of thermal expansion ($K^{-1}$) |
| $\gamma$ | Ratio of specific heats, $c^p$, $c^v$ (dimensionless) |
| $\Delta$ | Change in variable, final—initial |
| $\delta$ | Delta function (units vary) |
| $\in$ | Turbulent dissipation rate ($m^2/s^3$) |
| $\in$ | Void fraction (dimensionless) |
| $\eta$ | Effectiveness factor (dimensionless) |
| $\lambda$ | Wavelength (m, nm) |
| $\mu$ | Dynamic viscosity (cP, Pa-s) |
| $v$ | Kinematic viscosity ($m^2/s$) |
| $v'$, $v''$ | Stoichiometric coefficients for reactants, products (dimensionless) |
| $\rho$ | Density ($kg/m^3$) |
| $\sigma$ | Stefan-Boltzmann constant ($5.67 \times 10^8$ W/$m^2$-$K^4$) |
| $\sigma$ | Surface tension (kg/m, dyn/cm) |
| $\sigma_2$ | Scattering coefficient ($m^{-1}$) |
| $\sigma_k$ | Turbulent Prandtl number for $k$ |
| $\sigma_\varepsilon$ | Turbulent Prandtl number for $\varepsilon$ |
| $\overline{\overline{\tau}}$ | Stress tensor (Pa) |
| $\tau$ | Shear stress (Pa) |
| $\tau$ | Time scale, (s) |
| $\phi$ | Equivalence ratio (dimensionless) |
| $w$ | Specific dissipation rate ($s^{-1}$) |

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
