# Peer review of "The Use of Computational Fluid Dynamics for Assessing Flow-Induced Acoustics to Diagnose Lung Conditions"

_mca, doi:10.3390/mca28030064_

Round 1

Reviewer 1 Report

This manuscript proposes a method for diagnosing pulmonary diseases by analyzing vocal and cough changes of a patient. The method utilizes computational fluid dynamics to simulate the flow-induced acoustics in the lungs and differentiate between healthy and infected lungs. The study focuses on pneumonia as the model disease and aims to demonstrate the potential of this method to accurately represent the flow-induced acoustics and distinguish between healthy and infected lungs. The results show that the proposed method can detect the difference in vocalization between healthy and pneumonia-infected lungs, suggesting its potential for enhancing the understanding of flow-induced acoustics for different lung pathologies. The long-term objective is to use this method for remote diagnosis of pulmonary diseases via telemedicine.

The structure of the article is clear. However, method description and results presentation need major improvements. Please find comments below:

1. Did the authors perform mesh independence study? Although based on my experience, 10 million mesh cells appear to be large enough, a mesh independence study would be necessary before starting CFD model validation. Also, the numbers of mesh cells denoted in line 96 and 107 are different. Please confirm.

2. Sentences starting from line 120 to 123 are confusing to me. First, when the authors say "a second set of simulations", it implies that there is a first set of simulations, but I cannot find any information about it. Second, it's unclear how the boundary conditions are specified. Based on Table 1, respiratory bronchioles were selected as velocity inlet (this is also very questionable), but no information was given about how the inlet velocity/flow rate was determined and defined for each bronchiole. Since the lung geometry is subject-specific, which means the flow pattern is not symmetric at all, the flow rate at each bronchiole should also be different. Finally, can the authors explain the meaning and reason for "for the infected lung which is at a higher pressure"? How is the 1,4e6 Pa calculated, what pressure does this value represent?

3. Did the authors perform a time step independence study? Time step size of 0.1 seems a very large value to me, since turbulence is simulated in this study. Can the authors provide convergence results of their model?

4. Which model was used in this study, LES or k-epsilon? Line 112 mentioned that LES model is selected, but later in the section 2.3.2, k-epsilon model is described. Please confirm. If LES model is selected, then a time step of 0.1 second is obviously too large to get accurate results. 

5. Is the flow rate at mouth opening the same (which is PEFR) during the entire simulated period? Or is an exhalation profile used?

6. Please carefully revise section 2.3. Please provide descriptions or references for all the symbols in this section.

7. How valid is the assumption that the turbulence in respiratory system is isotropic?

8. Line 174, how to compute I, sound intensity?

9. How is Boussinesq Equation used in the model?

10. Page 8, please revise the figure number in the figure caption.

11. What the difference between the figure on page 9 and figures on page 10?

12. What is gauge pressure (line 242)?

Author Response

Good Day, 

Thank you for the review. Please find attached our response to the review. 

Regards,

Khanyisani

Reviewer 2 Report

1. What about the other parameters of adventitia? Are they the same as in the case of polyurethane or changed? Are the walls of the adventitia flexible or rigid in your model?

2. Do you use air with humidity or is it dry air? Also, do you account for compressibility of air in your model?

3. Lines 117-118: it is not velocity, it is volumetric flow rate.

4. Please elaborate more on the second set of simulations (lines 120-123), it is not entirely clear what you mean by that. Where do you apply that pressure boundary condition? Also, why healthy lungs need smaller pressure than infected lungs?

5. In lines 110-111 you wrote that the applied turbulence model is Large Eddy Simulation, but in lines 145-146 you wrote that it is k-epsilon model. Which one is correct? 

6. What excatly represents each dot on the scatter plots? Is it the surface acoustic power in different parts of the lungs?

7. If we know that volumetric flow rate is lower in the case of infected lungs, can't we use this parameter for diagnostics? 

Author Response

(The authors gave the same response as above.)

Round 2

Reviewer 1 Report

Thank the authors for providing such elaborate responses and most of my concerns have been well addressed.

However, the time step size used in this work still concerns me. Let's assume a trachea with a diameter of 20 mm. When the flowrate is 7.5 L/s, the average velocity in trachea is about 23.9 m/s. When time step size is 0.1 s, then the characteristic length simulated would be 2 m, which seems a large value if compared with lung size. It would make this work more solid if the authors can justify their choice of time step size, estimate the accuracy of current model setup, and show that 0.1 s generates accurate enough results.

Author Response

Good Day,

Thank you for the review. Please find attached our response for your review.

Kind Regards,

Khanyisani 

Reviewer 2 Report

The authors properly addressed the issues. The paper can be accepted.

Author Response

Good Day,

Thank you for reviewing the manuscript. 

Kind Regards,

Khanyisani 
